# The Pervasive Role of the miR-181 Family in Development, Neurodegeneration, and Cancer

**DOI:** 10.3390/ijms21062092

**Published:** 2020-03-18

**Authors:** Alessia Indrieri, Sabrina Carrella, Pietro Carotenuto, Sandro Banfi, Brunella Franco

**Affiliations:** 1Telethon Institute of Genetics and Medicine (TIGEM), Via Campi Flegrei 34, 80078 Pozzuoli, Naples, Italy; carrella@tigem.it (S.C.); p.carotenuto@tigem.it (P.C.); 2Medical Genetics, Department of Translational Medical Sciences, University of Naples “Federico II”, Via Sergio Pansini 5, 80131 Naples, Italy; 3Institute for Genetic and Biomedical Research (IRGB), National Research Council (CNR), 20090 Milan, Italy; 4Medical Genetics, Department of Precision Medicine, University of Campania “Luigi Vanvitelli”, 80138 Naples, Italy; 5The Institute of Cancer Research, Cancer Therapeutics Unit 15 Cotswold Road, Sutton, London SM2 5NG, UK

**Keywords:** microRNA, miR-181, embryo development, central nervous system development, Alzheimer’s disease, Parkinson’s disease, mitochondria, neurodegeneration, cancer

## Abstract

MicroRNAs (miRNAs) are small noncoding RNAs playing a fundamental role in the regulation of gene expression. Evidence accumulating in the past decades indicate that they are capable of simultaneously modulating diverse signaling pathways involved in a variety of pathophysiological processes. In the present review, we provide a comprehensive overview of the function of a highly conserved group of miRNAs, the miR-181 family, both in physiological as well as in pathological conditions. We summarize a large body of studies highlighting a role for this miRNA family in the regulation of key biological processes such as embryonic development, cell proliferation, apoptosis, autophagy, mitochondrial function, and immune response. Importantly, members of this family have been involved in many pathological processes underlying the most common neurodegenerative disorders as well as different solid tumors and hematological malignancies. The relevance of this miRNA family in the pathogenesis of these disorders and their possible influence on the severity of their manifestations will be discussed. A better understanding of the miR-181 family in pathological conditions may open new therapeutic avenues for devasting disorders such as neurodegenerative diseases and cancer.

## 1. Introduction

MicroRNAs (miRNAs) are endogenous small (20–22 nucleotides in length) non-coding RNAs, which play an important role in the regulation of gene expression. In the vast majority of cases miRNAs act by down-regulating the expression of their target genes. This action is mediated through imperfect binding to target sequences mostly located in 3′-untranslated regions [1]. An important role in target sequence recognition is played by the so-called “seed region,” i.e., a region of 7–8 nucleotides in the mature RNA sequence mostly located at positions 2–7 from of the miRNA 5’ region [2,3]. Depending on the recognition site, binding of miRNA-induced silencing complex to the cognate target can have different outcomes [4]. In the majority of cases, the binding is partially complementary to the target sites and leads to the repression of translation, whereas when it is fully complementary (mostly in plant cells), it leads to the degradation of the target transcript [1,2,5].

miRNAs can be organized in families composed of several members differing only for a few nucleotides outside the seed region. miRNA family members are usually clustered in specific chromosomal regions and can also be present in two or more copies [6]. Similar to other regulatory non-coding RNAs, microRNAs display discrete expression patterns ranging from restricted spatial-temporal localization to more widespread, if not ubiquitous, distributions [7,8]. A large body of experimental evidence demonstrates that miRNAs regulate fundamental biological processes and that their dysfunction can have deleterious effects both on developmental processes and on the homeostasis of mature tissues/organs [9].

Among the so far characterized miRNA families, we will focus on the miR-181 family, i.e., a group of highly conserved miRNAs of growing biomedical relevance. As we will discuss, the miR-181 family regulates many relevant biological processes such as cell proliferation, apoptosis, autophagy, mitochondrial function, and immune response [10,11,12,13,14,15]. Importantly, several studies have shown aberrant expression of these miRNAs in the most common neurodegenerative disorders (i.e., Alzheimer’s and Parkinson’s Diseases) [16,17,18,19,20,21,22,23] as well as in different solid tumors and hematological malignancies (see Table 1), where they act either as tumor suppressors or oncomirs (see Summary box). 

In the human genome, the miR-181 family is composed of four different 5p mature forms, namely (a) miR-181a-5p; (b) miR-181b-5p; (c) miR-181c-5p, and (d) miR-181d-5p (Figure 1A). These four mature miR-181 forms are generated from the following six different precursors (stem-loop) sequences: (a) pre-mir-181a1 and pre-mir-181a2 that give rise to two identical copies of mature miR-181a-5p; (b) pre-mir-181b1 and pre-mir-181b2 that generate two identical copies of mature miR-181b-5p; (c) pre-mir-181c that gives rise to the mature miR-181c-5p, and (d) pre-miR-181d that is processed to form the mature miR-181d-5p. It must be underlined that the above mir-181 precursors can also give rise to six different 3p mature sequences (Figure 1B), whose expression levels are however much lower, but not negligible, with respect to their cognate 5p forms [24]. miR-181 family members are distributed in three independent genomic clusters localized to three separate chromosomes, namely chromosomes 1, 9, and 19 [25] (Figure 1C). In particular, the cluster on chromosome 1 is composed of mir-181a1 and mir-181b1 (less than 100 base pairs (bp) apart) whereas the cluster on chromosome 9 is composed of mir-181a2 and mir-181b2, which are separated by about 1200 bp. Finally, the cluster on chromosome 19 is composed of mir-181c and mir-181d that are about 100 bp apart. The 5p mature forms produced by all these precursors share the same seed region and are therefore predicted to recognize a similar set of target genes (Figure 1A). Outside the seed regions, the four 5p mature sequence show between 1 to 4 base differences, with the pairs miR-181a/miR-181c and miR-181b/miR-181d displaying the highest similarity within each other (Figure 1A). This observation suggests that miR-181c and miR-181d may have derived from a duplication of miR-181a and miR-181b, respectively. On the other hand, the six 3p miR-181 family mature sequences show a higher extent of nucleotide difference with each other, even within the seed region (Figure 1B), and display lower expression levels vs. their related 5p counterpart [24]. Interestingly, despite the higher sequence heterogeneity, the 3p mature sequences show a similar trend of sequence similarity with the paralogs of the miR-181a/c pair being more similar between each other vs. the ones of the miR-181b/d pair.

MiR-181 family members are evolutionarily well conserved across all vertebrate species, particularly the paralogs “a” and “b”, whereas the paralogs “c” and “d” seem to have appeared more recently during evolution, likely via independent evolution of one of the a/b initial cluster, as also previously mentioned. As an example, in the medaka fish *(Oryzias latipes)* genome, four clusters are present all of which characterized by the presence of a/b mature sequences [26]. 

Experimental evidence obtained in different laboratories demonstrates that the miR-181 family is mainly expressed in neuronal, blood and lymphoid tissues. This is particularly true for miR-181a and miR-181b (miR-181a/b) while miR-181c and miR-181d display a more widespread expression as also evident after analysis with the tool available at http://ngs.ym.edu.tw/ym500v2/knownmir.php. In the particular analysis of genetic animal models harboring a targeted deletion of the miR-181a/b-1 loci revealed that this cluster accounts for most of the expression of the mature miR-181a/b in the brain and retina [24,27,28].

The sections below summarize our current knowledge on the pervasive role of this miRNA family in development and differentiation, neurodegenerative diseases and cancer. (Table 1).

## 2. The miR-181 Family in Development and Differentiation

miRNAs have been found to play relevant roles in several fundamental developmental processes among which gametogenesis, preimplantation and early embryonic development, neurogenesis and somitogenesis [8,9,29,30,106,107]. As a consequence, their expression during development needs to be tightly regulated in time and space. The expression of miR-181 family members represents a paradigmatic example in this respect, as reported below. 

### 2.1. Early Embryo Development

The importance of tight expression regulation of miR-181a/b during development is already evident at embryo implantation. It is known that the cytokine leukemia inhibitory factor (*LIF*) is essential for embryo implantation and is transiently expressed in the uterine endometrial glands at days 4 post-fertilization (embryonic day 4, E4) in mice. Analysis of miRNA expression profiles in mouse uterus revealed that all miR-181 family members are downregulated specifically at E4 vs. non-pregnant mice and that their normal levels are restored at E6 and E7 [29]. In utero overexpression of miR-181 family members indicate that they play an inhibitory role in the regulation of embryo implantation via direct targeting of *LIF*. Interestingly, it was shown that the miR-181a1/b1 and miR-181a2/b2 genomic clusters are transcriptionally activated by the Empty Spiracles Homeobox 2 (*Emx2*) transcription factor whose levels, similar to miR-181a/b during pregnancy, are decreased at E4 but recover rapidly at E7. Chromatin immunoprecipitation assays confirmed the association of endogenous Emx2 and the chromatin fragments corresponding to the Emx2-binding elements in miR-181a1/b1 and miR-181a2/b2 promoters. Altogether these data highlight the importance of the Emx2–miR-181a/b–LIF axis in the regulation of embryo implantation [29].

Moreover, miR-181a/b overexpression in medaka fish results in lethality at gastrulation with the relatively few surviving embryos showing a small body size and head defects, with enlargement of the otic vesicle, and in some cases the complete absence of eye structures [26].

### 2.2. Late Embryo Development

Later in development, the tight regulation of miR-181 family members is of fundamental importance during interdigital tissue remodeling in the embryonic limb [30]. The development of the limb bud is accompanied by precise patterns of massive mesodermal programmed cell death required to eliminate undifferentiated cells which are needed only for a limited period of limb development. This morphogenetic process is finely regulated by transforming growth factor-β (TGF-β) signaling. Analysis of miRNA expression, by next-generation sequencing, highlighted very high levels of miR-181a/b in chick embryo interdigits. miR-181a and miR-181b together constitute over 10% of the total miRNA content at the onset and 30% at the peak of interdigital cell death. Interestingly the latter miRNAs are involved in the control of apoptosis [10,108] and the regulation of TGF-β signaling [35]. These data were confirmed in embryonic duck limb during tissue remodeling. Quantitative polymerase chain reaction experiments showed increased expression levels of miR-181a at stages of maximum apoptosis and a drop at post-apoptosis stages compared with their expression levels before the onset of tissue regression [30]. At the same time, during embryonic limb formation, the development of cartilage begins with the condensation of mesenchymal stem cells (MSC) followed by differentiation toward the chondrocyte lineage [109]. A comprehensive miRNA-array analysis was performed on five maturation stages during in vitro-induced differentiation of human MSC into chondrocytes under the classical TGF-β-driven protocol, which closely recapitulates the fundamental stages of embryonal chondrocyte development, namely MSC, prechondrocytes, chondroblasts, chondrocytes, and hypertrophic chondrocytes [32]. The most interesting differential expression during maturation stages was detected for miR-181 family members. In particular, they all displayed low levels in prechondrocytes and were significantly enhanced in chondroblasts, with miR-181b raising further in chondrocytes, and miR-181a in hypertrophic chondrocytes. Accordingly, miR-181a was associated with osteoblast development and endochondral ossification of mouse calvaria and tibia [31] and its up-regulation was observed in vivo during hypertrophic chondrocyte differentiation [33,110].

During development, the miRNA-181 family is also highly expressed in thymus, lung, bone marrow, and spleen [111], having a role in the differentiation of specific cell types in these tissues. In particular, the miR-181a1/b1 cluster is highly expressed in the thymus, which is a highly proliferative tissue with large metabolic requirements. Activation of the phosphoinositide 3-kinase (PI3K) pathway provides a potent stimulus for proliferation and growth during T-cell and Natural Killer (NK)T-cell development in the thymus. Using an in silico approach to search for miRNAs with the capacity to regulate the PI3K pathway, four predicted miR-181 sites were discovered in the 3’ UTR of the phosphatase and tensin homolog (*PTEN*) transcript [12]. PTEN, a 3’-phosphate lipid phosphatase, is the main negative regulator of the PI3K pathway, which dephosphorylates Phosphatidylinositol (3,4,5)-trisphosphate (PtdIns(3,4,5)P3), abbreviated PIP3, to Phosphatidylinositol 4,5-bisphosphate or PtdIns(4,5)P2, also known simply as PIP2, thereby preventing its accumulation [112]. To test the ability of the miR-181 family to regulate PTEN in vivo, mice deficient in each of the three miR-181 clusters were generated. Thymocytes from miR-181a1/b1-deficient mice displayed elevated *PTEN* levels and a concomitant reduction in PI3K signaling, inducing a major metabolic reprogramming. The PI3K-directed metabolic changes are required to support appropriate thymocyte and NKT cell development and miR-181a1/b1-deficient thymocytes recapitulate numerous phenotypes displayed in PI3K-deficient animals, consistent with a role of miR-181 in regulating the PI3K/PTEN signaling axis. Therefore, miR-181a1/b1-deficient cells fail to fully reach the biosynthetic potential of normal proliferating thymocytes. Moreover, miR-181 deficiency was accompanied to developmental block and reduced proliferation during the early stages of NKT cell development [12]. In addition, miR-181a1/b1; miR-181a2/b2 double knockout mice showed reduced survival and a significant reduction in body size, while triple knockout mice were never obtained, suggesting that complete deficiency of the miR-181 family is lethal [12]. These data indicate that the miR-181 family plays an essential role in growth and/or development in a dose-dependent manner [113].

### 2.3. Central Nervous System (CNS) Development

The ability of the miR-181 family to act as a metabolic rheostat turns is fundamental also in the brain. During brain development, PTEN negatively regulates neural stem cell self-renewal [114]. In cultured embryonic striatal stem cell (ESSC), miR-181c is up-regulated upon LIF+IGF-1 treatment [34]. This up-regulation could push the cell fate discrimination to an off-balanced state, leading to an enhanced self-renewal of the ESSC pool. This process occurs by abrogating the lineage commitment, probably via miR-181 targeting of *PTEN* as well as of other phosphatases involved in neuronal differentiation (protein tyrosine phosphatase non-receptor type 11 PTPN11, protein tyrosine phosphatase non-receptor type 22 PTPN22 and dual-specificity phosphatase 6 Dusp6) [34].

The role of a miR-181 family member in the CNS is also exemplified when analyzing retinal development. miR-181a and miR-181b are expressed in the Inner Nuclear Layer and Ganglion Cell Layer of the vertebrate retina [26,115], and at very low levels in the Outer Nuclear Layer [28,116]. In vertebrate models, such as medaka fish, both miR-181a and miR-181b are expressed in differentiating amacrine and ganglion cells of the developing retina and in different regions of the CNS, including the pretectal and tectal areas, which are visual areas in vertebrates [26]. As previously mentioned, in the medaka fish genome, the miR-181 family is organized in four different clusters only containing miR-181a and miR-181b. For this reason, the loss of miR-181a/b mature forms in medaka fish represents the loss of the entire miR-181 family. Morpholino (MO)-mediated combined knockdown of miR-181a/b in medaka fish leads to a specific retinal phenotype characterized by the reduction of Inner Plexiform Layer (IPL) thickness, without any apparent reduction in the number of retinal cells [26]. This leads to a failure in the consolidation of amacrine cell processes into axons and a delay of the growth of retinal ganglion cell (RGC) axons, without altering the timing of onset and progression of retinal cell differentiation. The above alterations translate into impairment of retinal circuit assembly and to visual function defects [26]. These phenotypes are mainly mediated by the miR-181a/b direct targeting of Extracellular Signal-Regulated Kinase 2 (*ERK2*), a kinase member of the MAPK signaling cascade, whose dysregulation alters cytoskeleton remodeling in axonal growth cone preventing axonal elongation [26]. Moreover, in the retina, TGF-β acts on the post-transcriptional maturation of miR-181a/b increasing the levels of their mature forms [35]. As a consequence, the TGF-β signaling controls the miR-181/ERK regulatory network, which in turn strengthens the TGF-β-mediated regulation of RhoA degradation [35,117]. These data highlight the miR-181a/b as a key node in the interplay between TGF-β and MAPK/ERK within the functional pathways that control retinal axon specification and growth (Figure 2). Interestingly, this role is dose-dependent because the IPL phenotype observed in the double-morphant embryos (MO-miR-181a/b), presents similar onset and progression but is considerably stronger and more highly penetrant when compared to the single depletion of miR-181a and miR-181b [26]. In agreement with a dose-dependent effect and functional redundancy of miR-181 family members, the disruption of the miR-181a1/b1 cluster only did not cause abnormalities in retinal morphology nor alteration of retinal function in the mice [28]. These data on retinal tissues support the concept that the ablation of all members of a miRNA family is necessary to reveal the complete range of miRNA functions in animal models. This is also demonstrated in mouse models in which the ablation of more than one miR-181 cluster leads to a more severe phenotype not observed in single cluster deficient animals [113]. 

Overall, all of the above observations strengthen the importance of miR-181 family members in the modulation of tissue differentiation and remodeling. In particular, the tight regulation of expression of miR-181 family members in early phases of embryo development illustrates the impact of precise miRNA function regulation in specific and relevant life processes. Moreover, the role of these miRNAs in the immune system provides a paradigmatic example of how a miRNA family can act as a cellular metabolic rheostat during development [12]. 

## 3. The miR-181 Family in Neurodegenerative Diseases

miRNAs show a variety of important physiological functions in the CNS and perturbations of their expression profiles or those of their target genes have been associated with pathological processes affecting this system, including neurodegeneration [118,119]. The miR-181 family is highly expressed in different regions of the CNS and a functional role for the miR-181 family in more specific neurological processes such as neurotrophin signaling pathway, axon guidance, immunity, and mitochondrial-related pathways is also emerging [12,26,28,35,120]. Moreover, the deregulation of miR-181 family members has been reported in patients and animal models of neurodegenerative disorders such as Alzheimer’s Disease (AD) [11,16,17,18,19,20,21], Parkinson’s Disease (PD) [22,23], and multiple sclerosis [121,122].

Altogether these data suggest that these miRNAs could have direct relevance in the pathogenesis of neurodegenerative disorders and/or can influence the severity of their manifestations as detailed below.

### 3.1. Alzheimer Disease (AD)

miR-181c was shown to be down-regulated in the brain, cerebrospinal fluid (CSF) and blood of AD patients [17,18,19,21]. In addition, loss of this miRNA increases the levels of Serine palmitoyltransferase (SPT), the first rate-limiting enzyme in the de novo ceramide synthesis, which in turn increases the levels of the amyloid precursor protein (A*β*) [123]. Geekiyanage and colleagues showed that serine palmitoyltransferase long chain 1 (*SPTLC1*), encoding a component of the SPT heterodimer, is a direct miR-181c target. Down-regulation of *SPTLC1* by miR-181c overexpression reduced A*β* levels in primary astrocytes derived from transgenic mice expressing the human *APP* (Amyloid Precursor Protein) Swedish mutation [123]. These data indicate an important role of miR-181c in AD and propose this miRNA as a potential therapeutic target for this condition. 

However, the role of the other mir-181 family members in AD is more controversial, as independent reports show contrasting results concerning their differential expression in patients. In 2008, Cogswell and colleagues showed that miR-181a is down-regulated in CSF samples of AD patients compared to healthy controls [17], whereas two different reports showed that both miR-181a and miR-181b are up-regulated in AD patients’ blood samples [16,20]. Moreover, one of the latter studies described miR-181a up-regulation in mild cognitive impairment patients before conversion to AD. Notably, miR-181a upregulation also correlates with AD hallmarks such as increased CSF Aβ concentration, hippocampal atrophy, and disconnections in white matter brain regions [16]. 

Up-regulation of miR-181a was also reported in dorsal and ventral hippocampal regions of twelve-month old 3xTg-AD mice [11]. The authors also showed that the levels of both c-Fos and Sirt-1, which are important regulators of synaptic plasticity and memory [124,125], were significantly decreased in the ventral hippocampus of these mice. In addition, overexpression of miR-181a in SH-SY5Y cells decreased c-Fos and Sirt-1, suggesting that miR-181 regulates the expression of these two proteins. Notably, *c-Fos* and *Sirt-1* were identified as direct miR-181a targets in other studies [126,127]. Interestingly, Sirt-1 is connected to AD by different mechanisms, including those involved in APP processing, neuronal inflammation and degeneration, and mitochondrial dysfunction. In these contexts, the action of Sirt-1 may suppress AD pathology by diverse pathways [128]. Finally, very recently, Rodriguez-Ortiz et al. showed that inhibition of miR-181a effectively reverses Aβ-induced impairments in plasticity and memory deficits in primary hippocampal cultures and 3xTg-AD mice [36].

Nevertheless, additional efforts are necessary to clarify the role of the miR-181 family in AD and to explain the discrepancies of miR-181 expression levels in AD patients between different studies.

### 3.2. Parkinson’s Diseases (PD)

A possible role of the mir-181 family in PD has also been proposed. mir-181a and b are expressed in the Substantia Nigra and striatum [27,129] and were found to modulate the expression of mitochondrial-dependent cell death and autophagy-related genes (i.e., B-cell lymphoma 2 (*BCL2*), Myeloid Cell Leukemia 1, (*MCL1*), X-linked inhibitor of apoptosis protein (*XIAP*), autophagy-related gene (*ATG*) *5, ATG7*) [10,14,15,130]. Moreover, *PARK2* (Parkin), responsible for a monogenic form of PD and involved in the regulation of mitochondrial quality control and mitophagy [131], is a miR-181a direct target [37]. Notably, the latter pathways have been linked to PD pathogenesis and progression [132,133,134,135]. Moreover, mitochondrial dysfunction and overproduction of reactive oxygen species (ROS) have been suggested to play a major role in PD pathogenesis, and many genes associated with PD encode proteins that impact on mitochondrial function, clearance and oxidative stress [133]. Interestingly, a direct role of mir-181a and b in mitochondrial biogenesis and function has also been described [28]. Indeed, these two miRNAs control both *NRF1* (Nuclear respiratory factor 1) and *PPARGC1A* (PPARG coactivator 1 alpha, Pgc1a), master regulators of mitochondrial biogenesis, directly and indirectly, respectively. Moreover, mir-181a/b also control Cytochrome c oxidase copper chaperone (*COX11*) and coenzyme Q10B (*COQ10B*), which are involved in mitochondrial respiratory chain (MRC) assembly, and peroxiredoxin 3 (*PRDX3*), an important mitochondrial ROS scavenger [28]. Overall, these data indicate that the mir-181 family plays a pivotal role in mitochondrial function and point to miR-181a/b as possible hubs in the gene pathways underlying mitochondrial homeostasis in both physiological and pathological conditions (Figure 3).

Finally, differential expression of miR-181 family components were reported in PD patients: miR-181a resulted up-regulated in the Prefrontal Cortex of PD patients [23] whereas low expression levels of miR-181a were reported in the blood of PD patients [22]. As also discussed for AD, more efforts are necessary to better clarify this issue considering the discrepancies observed in miR-181 expression levels, and several factors such as accuracy of the diagnosis, gender, medication history, tissues analyzed, and methodologies used in the analysis (including samples acquisition and processing) should be carefully evaluated.

### 3.3. The miR-181 Family As A Therapeutic Target in Neurodegenerative Diseases

As discussed above, miR-181 family members simultaneously modulate in the CNS multiple molecular pathways involved in disease pathogenesis and progression. On this basis, they may also represent new therapeutic targets for neurodegenerative diseases.

Interestingly, independent reports indicate that miR-181 down-regulation could be protective against neuronal cell death in different in vitro and in vivo models. Indeed, inhibition of miR-181a by antagomir was shown to effectively decrease brain infarction in mouse models of stroke by increasing Bcl-2 and Xiap expression [38]. In particular, post-stroke treatment with miR-181a antagomir also reduces long term neurobehavioral deficits indicating that miR-181a inhibition has neuroprotective effects against ischemic neuronal damage and neurological impairment in mice [38].

Recently, a protective effect of miR-181a and b down-regulation has been also reported in different models of mitochondrial diseases (MD) characterized by neuronal degeneration [28]. The authors showed that miR-181a and b control global regulation of mitochondrial turnover in the CNS through coordinated activation of mitochondrial biogenesis and mitophagy [28]. They demonstrated that the silencing of these two miRNAs significantly rescues neurodegeneration in two fish models of Microphthalmia with linear skin defects syndrome (MLS; MIM309801 [136]), a rare neurodevelopmental disorder due to mutations in players of the MRC Complex III and IV [137,138,139]. Moreover, they tested the effects of genetic inactivation of miR-181a and b in a chemical (rotenone-induced [140]) and genetic (*Ndufs4^-/-^* [141]) murine models of Leber hereditary optic neuropathy (LHON, MIM535000 [142]), a mitochondrial disorder characterized by degeneration of RGCs and loss of central vision. Notably, miR-181a and b down-regulation exerted a protective effect in all tested models, regardless of the underlying etiopathogenetic genetic events thus highlighting these two miRNAs as potential, gene-independent, therapeutic targets for MDs characterized by neuronal degeneration [28]. 

In conclusion, all of the above observations indicate that the miR-181 family could play a key role in the pathogenesis of different neurodegenerative disorders. Moreover, several studies suggest that these miRNAs also represent new therapeutic targets and can serve as diagnostic or prognostic biomarkers in these disorders.

## 4. The miR-181 Family in Cancer

Several studies have demonstrated that miR-181 family members have a pivotal role in cancer, acting either as tumor suppressors or oncomirs and modulating a wide range of mRNA targets belonging to the main cancer-related pathways. Recent findings show the implication of miR-181 family members in different solid tumors and hematological malignancies. The following paragraphs describe the links of each member with target genes and cancer-related pathways (for a summary, see Table 2).

### 4.1. Solid Tumors 

#### 4.1.1. Hepatocellular Carcinoma

Wang’s group reported a genome-wide screening to identify microRNAs associated with hepatic cancer stem cells. They first reported that all four mature miR-181 family members (miR-181a, miR-181b, miR-181c, miR-181d) were significantly increased in twenty Hepatocellular carcinoma (HCC) cases as well as in HCC stem cells and progenitors. They demonstrated that all four miR-181 family members play important roles in the maintenance of HCC stem cells by down-regulating two hepatic transcriptional regulators, namely *CDX2* and *GATA6*, and the Wnt signaling inhibitor *NLK* [39].

The association between miR-181 members and the Wnt pathway has been further demonstrated by the same authors showing that the Wnt/beta-catenin signaling is a major transcriptional regulator of miR-181 expression in HCC. They also provided evidence of in vivo binding between the Tcf4/β-catenin complex and the TCF/LEF binding elements in the promoter region of the miR-181a-2/miR-181b-2 transcripts [64].

Wang’s group also reported a significant association of miR-181b and miR-181d with hepatocarcinogenesis using tumorigenic in vitro assays in an appropriate mouse model. In particular, they demonstrated that the TGFβ pathway modulates the transcription of *miR-181b* which in turn can promote tumorigenicity and resistance of HCC cells to the anti-cancer drug doxorubicin by targeting the tumor suppressor *TIMP3* [40].

The implication of miR-181a, b, and d in chemically-induced hepatocarcinogenesis has been assessed also by Song and colleagues, who showed that this miRNA family plays a critical role in chemically-induced hepatocarcinogenesis by targeting *MKP-5* and regulating the p38 MAPK activation [41]. In a recent study aimed at analyzing the functional roles of the miRNome in intra-hepatic and extra-hepatic bile duct cancer, no impact on cell tumor growth was observed after the miR-181 inhibition of miR-181 family members [65]. 

#### 4.1.2. Pancreatic Carcinoma (PDAC)

Accumulating evidence highlighted the role of miR-181a and b as “oncomiRs” in pancreatic cancer. In particular, miR-181a has been described to promote pancreatic cancer invasion and progression by targeting the tumor suppressor genes *PTEN* and *MAP2K4* [48]. Furthermore, the expression levels of plasma miR-181a and b were found to be significantly increased in pancreatic cancer patients compared to patients with pancreatitis and normal controls [66].

An expression profiling in plasma of 54 patients revealed that circulating miR-181a-5p was significantly down-regulated after treatment with FOLFIRINOX (5-fluorouracil, oxaliplatin, and irinotecan). This down-regulation correlated with improved survival of the patients analyzed. The authors also showed that ectopic over-expression of miR-181a increases the proliferation of *PDAC* cells by targeting the *ATM* gene, while the inhibition of miR-181a in combination with oxaliplatin treatment increased DNA damage and decreased cell viability. The clinical data, together with in vitro findings, strongly suggest the use of miR-181a as a biomarker to select patients responsive to FOLFIRINOX therapy [49,67].

#### 4.1.3. Oral Cancer (OC)

Oral cancer is one of the most prevalent cancers worldwide. A recent study, involving the analysis of 36 differentially expressed miRNAs in high-grade oral cancers, revealed that miR-181c-5p is associated with a real prognostic significance in terms of patient overall survival [68]. Furthermore, over-expression of miR-181 was correlated with lymph node metastasis, vascular invasion, and poor survival in OC by Lin et al [69]. Functional assays revealed that the over-expression of miR-181 would enhance cell migration and invasion but not the ability of anchorage-independent growth of OC cells [69].

#### 4.1.4. Colorectal Carcinoma (CRC)

Several studies reported that miR-181a and b are abnormally expressed in CRC tissue and that their level of expression may be associated with unfavorable clinical outcome [44,70,71,72,73,74,75,76]. In recent seminal work, performing a comprehensive meta-analysis in a selected collection of 1017 patients, Peng et al., showed that increased expression of miR-181 family members could be used as a prognostic biomarker [77]. In addition, they also explored the function of miR-181 members through an integrative bioinformatics analysis aimed at the identification of putative miR-181a and b target genes. The predicted targets were found to be enriched in pathways closely related to CRC as MAPK, FoxO, VEGF, HIF-1, PI3K-Akt, mTOR, and in central carbon metabolism. Previous studies have indicated that miR-181a plays a key role in promoting angiogenesis in CRC by targeting *SRCIN1* to promote the VEGF signaling pathway [42]. Overexpression of miR-181a was found to effectively promote CRC cell growth rate, migration and invasion in vitro and tumor growth and liver metastasis in vivo as described by Ji et al., suggesting that miR-181a plays a critical oncogenic role in the invasive and/or metastatic potential of CRC [44]. The tumor-promoting function of miR-181a is exerted through repression of its downstream target gene *WIF-1* (Wnt inhibitory factor-1), an inhibitor of the Wnt signaling pathway [44].

In contrast with previous studies showing that the expression levels of miR-181a are positively correlated to a poor prognosis and decreased survival time, Pichler et al. demonstrated a significant association between low expression of miR-181a and poor survival in 80 CRC patients [73]. The same evidence was obtained by Zhang et al. when analyzing the expression of miR-181b in 97 human CRC samples. The authors reported an association between the down-regulation of miR-181b and tumor progression. They also showed that overexpression of miR-181b induced apoptosis in CRC cells and tumor cell growth inhibition by direct targeting of the *RASSF1A* gene [43].

#### 4.1.5. Non-Small Cell Lung Cancer (NSCLC)

A recent work by Pop-Bica et al. reported a meta-analysis based on 2653 cancer patients to investigate the role of miR-181 family members in patient outcomes. The analysis indicated that miR-181a expression levels significantly correlated with survival in patients with non-small-cell lung cancer [78]. A global expression profile of miR-181 family members in non-small cell lung cancer (NSCLC), carried out by Huang et al., has provided evidence that miR-181 members were down-regulated both in NSCLC tissues and cell lines. The same authors showed that the overexpression of these miRNAs inhibited NSCLC cell proliferation, migration, and invasion and promoted cell apoptosis through *BCL2* targeting [108].

In NSCLC cell lines, it was also demonstrated that miR-181b overexpression sensitized multidrug-resistant cells to cisplatin-induced apoptosis by targeting *BCL2* [45]. Moreover, miR-181b has also been shown to inhibit migration and invasion of NSCLC by directly targeted high-mobility group box-1 (*HMGB1*) [46].

In a recent study, the involvement of miR-181a-5p in NSCLC progression has been elucidated by providing also evidence that this miRNA inhibits NSCLC cell proliferation and migration by targeting *VCAM-1*. The authors also showed that the interplay between miR-181a and VCAM-1 is regulated by the pro-inflammatory cytokine Interleukin (IL)-17 and the NF-κB pathway [47].

#### 4.1.6. Ovarian Cancers

The relevance of miR-181 family members in ovarian cancer is mainly supported by the genomic location of miR-181a1b1 in the 1q31.3–1q32.1 chromosomal region, frequently involved in ovarian cancers [79].

A critical role in ovarian cancer progression for miR-181a has been highlighted by the group of Parikh et al. who first explored the functional and clinical relevance of miR-181 members. In a patient cohort of high-grade ovarian cancer, the expression levels of miR-181a and b were found to be correlated with poor outcome. They also investigated the functional role of miR-181a by reporting in vitro and in vivo data demonstrating that miR-181a increases migration and invasion by regulating epithelial-mesenchymal transition (EMT). The clinical relevance was further assessed by identifying *SMAD7* as the functional target driving miR-181a-mediated TGF-β pathway activation [50].

Over-expression of miR-181a in epithelial ovarian cancers was further confirmed by Chong et al. in a Korean cohort of patients [80]. However, in contrast to the data previously described, more recent findings indicate that miR-181 members act as tumor suppressors and play a substantial role in inhibiting tumorigenesis and reversing the metastasis of ovarian cancer by targeting the *RTKN2* gene and suppressing the NF-κB signaling pathway in vitro [51].

#### 4.1.7. Prostate Cancer (PCa)

The findings by He et al., indicating that miRNA-181b may act as an oncomiR in prostate cancer [13], were further confirmed by Tong at al., who showed that miR-181 members are significantly up-regulated in prostate cancers. The authors also demonstrated that miR-181a promotes tumor growth in vitro and in vivo and identified *DAX-1*, a gene regulating androgen receptor signaling, as a direct target of miR-181a in prostate cancer cells [52]. The oncogenic roles of miR-181a were also confirmed by a more recent work reporting that miR-181a is highly correlated with the pathogenesis of PCa. The oncogenic activity of miR-181a has been functionally linked to the suppression of the Smad repressor TGIF2 and to the promotion of the EMT process in PCa cells [53]. More recently, according to the hypothesis that miR-181a acts as oncomiR, it has been shown that miR-181a is up-regulated in advanced stages of prostate cancer and it is involved in drug resistance [81].

In contrast with previous results, recently, Lin et al. described a tumor suppressor role of miR-181b showing that this miRNA can inhibit cell growth and enhance apoptosis in a prostate cancer cell line. By gene set enrichment analysis, they also identified a molecular signature of genes down-regulated by miR-181b. The most statistically relevant pathways are MYC, G2M checkpoint, E2F, Mitotic spindle, MTORC1 signaling, Oxidative phosphorylation, EMT, Fatty acid metabolism, Glycolysis, Adipogenesis [82].

Recently, it has been found that overexpression of miR-181a significantly inhibited cell proliferation by inducing G1-phase cell cycle arrest. Furthermore, miR-181a was down-regulated in PCa samples, strongly suggesting that miR-181a may act as a tumor suppressor in PCa.

#### 4.1.8. Breast Cancer

The oncogenic role of miR-181 has been assessed in breast cancer by several studies [54,83]. The association of miR-181a and b with breast cancer aggressiveness has been functionally correlated to the inhibition of *ATM* (ataxia telangiectasia), *DDR* (DNA damage response) transcripts and the repair of DNA damage pathways [83]. Furthermore, Yoo et al. reported that miR-181b can increase invasiveness and induce epithelial-mesenchymal transition in breast cancer cells by targeting *YWHAG* (tyrosine 3-monooxygenase/tryptophan 5-monooxygenase activation protein γ) expression [54]. The tumor-promoting function of miR-181b was further demonstrated by Zheng et al. who confirmed the up-regulation of miR-181b in breast cancer cell lines and patients and its oncogenetic role in promoting cell proliferation and migration in addition to inducing chemoresistance by targeting the mitochondrial-derived apoptogenic proteins Bim [55]. Similarly, Neel et al. reported that miR-181s exhibits pro-migratory and pro-invasive effects in breast cancer cells [84]. In addition, the serum levels of miR-181 members were markedly decreased in patients with early-stage breast cancer following surgical resection [85]. The molecular mechanisms underlying miR-181-mediated effect in malignant progression of breast cancer have been elucidated and associated with a novel target, the tumor suppressor *SPRY4* (Protein sprouty homolog 4) [56].

#### 4.1.9. Brain Cancers

##### Astrocytoma

Astrocytoma, the most common neuroepithelial cancer, represents the majority of malignant brain tumors in humans. Zhi et al. provided evidence indicating that miR-181b acts as a tumor suppressor in astrocytomas. The down-regulation of miR-181b in astrocytomas has been associated with a poor prognosis and the overexpression of miR-181b inhibits astrocytoma cell proliferation, migration, and invasion and promotes apoptosis. It has been shown that miR-181b exerts its function by targeting *NOVA1*, which belongs to the Nova family of neuron-specific RNA-binding proteins [57].

##### Glioma and Glioblastoma

Glioma and Glioblastoma are the most common and aggressive types of adult brain tumors. Several reports showed that miR-181a and b act as tumor suppressors by inhibiting tumor growth and invasion and by inducing apoptosis in glioma cells [86]. Wang’s group assessed the role of miR181 family members in glioma, showing that the expression levels of these miRNAs were down-regulated in glioma patients. The inhibition of tumor cell growth by miR-181 family members has been confirmed and its functional role was linked to the targeting of cyclin B1 [58].

Expression analysis on both glioblastoma tissues and glioblastoma cell lines revealed that miR-181a, b, and c are down-regulated in glioblastoma [87]. In support of this observation, another study revealed these miRNAs were significantly down-regulated in glioblastomas compared to adult brain tissues and that the lower levels of miR-181b and c in glioblastomas were positively correlated with response to chemoradiotherapy in glioblastoma patients [88].

More recent studies, carried out in two independent cohorts comprising nearly 700 glioblastoma patients, revealed that patients with high miR-181 expression levels had longer overall survival and that these miRNAs may act as a tumor suppressor by targeting the NF-κB and EMT pathways. miR-181b has been shown to be the most active in modulating the above-cited pathways by targeting *KPNA4* expression both in vitro and in vivo [59,89].

##### Neuroblastoma (NB)

As reported by Gibert et al., miR-181a and b are up-regulated and associated with poor prognosis in Neuroblastoma (NB), the most common extracranial pediatric solid tumor [60,90]. Interestingly, expression levels of miR-181c and d showed no association with NB prognosis. Direct regulation of the tumor suppressor *CDON* (cell-adhesion molecule-related/down-regulated by oncogenes) by miR-181 family members in NB cells was demonstrated [60]. As CDON has pro-apoptotic activity, it has been suggested that miR-181 members may act as oncomiRs by targeting CDON in NB pathogenesis [60]. Furthermore, a direct correlation between *MYCN* oncogene amplification and miR-181a and b expression, highlighted by Chen et al., has not been confirmed by other authors in NB patients. On the other hand, it has been shown in NB cell lines, that miR-181a and b were highly expressed in *MYCN* amplified cells [60,90].

As previously discussed, miR-181a has been identified as a modulator of mitophagy repression in NB cell lines and as an inducer of apoptosis by direct targeting *PARK2* (Parkin RBR E3 ubiquitin-protein ligase) [37].

The functional role of miR-181c in NB has also been assessed, revealing that this miRNA is down-regulated in metastatic NB tissues and inhibits NB cell proliferation, migration, and invasion. Functional studies have suggested that miR-181c acts as a tumor suppressor by negatively regulating *SMAD7*, an inhibitory factor of the TGF-β1 pathway in NB [61].

### 4.2. Hematological Cancers

The aberrant expression of the miR-181 family has been observed in several hematological cancers, ranging from leukemia (chronic lymphocytic leukemia and acute myeloid leukemia) to Multiple Myeloma. Several members of the miR-181 family, mostly miR-181a and b, have been demonstrated to be functionally associated with the etiopathogenesis of hematological cancers, mainly by regulating the differentiation and development of immune cells, including natural killer cells and B and T lymphocytes, myeloid lineages, and erythroid and megakaryocytic cells. Accumulating evidence suggests that members of the miR-181 family may be a target for the treatment of hematological cancers and a useful biomarker in precision medicine [91].

#### 4.2.1. Leukemia

Pekarsky et al. [92] identified the aberrant expression of several members of the miR-181 family (mostly miR-181a and b) in patients within B-cell chronic lymphocytic leukemia (B-CLL), the most common form of human leukemia. According to the data reported by Visone et al., in Chronic myeloid leukemia [91], miR-181b regulates the expression of the *Tcl1*, *Mcl1*, and *BCL2* oncogenes, all of which are involved in the pathogenesis of the aggressive form of B-CLL. Calin et al. [63] further demonstrated that the high expression levels of the *Tcl1* oncogene in patients with B-CLL were a consequence of the down-regulation of miR-181b, which directly targets this oncogene.

Recently published studies provided evidence that miR-181b is involved in the B cell receptor (BCR) pathway and could contribute to the regulation BCR signaling in malignant B-CLL cells [93].

In chronic myeloid leukemia (CML), the expression levels of miR-181a and b were found to be significantly reduced in CML patients and the CML K562 cell line, as reported by Wang et al. [94], and Visone et al. [91] suggesting a role of tumor suppressor role for these miRNAs. Moreover, Wang et al. [94], demonstrated that ectopic overexpression of miR-181a in the CML K562 cells inhibits cell growth and/or to induce cells apoptosis and differentiation. They also showed that overexpression of miR-181a enhances the chemotherapeutic sensitivity of CML K562 cells to imatinib, suggesting a potential role of miR-181a as a predictive biomarker in the imatinib treatment. [94].

Analysis of microRNA expression profiles in patients with acute myeloid leukemia (AML) revealed that down-regulation of the miR-181 family contributes to an aggressive leukemia phenotype through mechanisms associated with the activation of pathways controlled by toll-like receptors and interleukin-1β [95]. Guo et al. found several members of the miR-181 family (miR-181a-5p, miR-181d-5p, miR-181b-5p) to be over-expressed in patients who progressed from myelodysplastic syndromes to AML [96].

Several studies investigated the role of miR-181 family members in AML was correlated with longer patient survival [97,98,99,100]. In particular, Li et al. [101] reported that increased expression of miR-181 was associated with a better prognosis. These results were confirmed by Butrym et al. who showed that low miR-181 expression levels were associated with complete remission and prolonged survival in older AML patients after treatment with azacitidine indicating that these miRNAs regulate the response to azacitidine [99]. The involvement of miR181a in the modulation of the cell cycle by regulating p27^Kip1^ expression in AML cells has been described [101], as well as the link between the down-regulation of miR181b and cell cycle control [102].

#### 4.2.2. Lymphoma

The role of miR-181a and b in Lymphoid cell development has been reviewed by Guarini et al. [62], and specific expression profiles linked to the different steps of the maturation/differentiation process largely documented [103,104]. In a recent study, it has been shown that miR-181b directly down-regulates the expression of *FAMLF* (Familial acute myelogenous leukemia related factor) and inhibits cell viability by binding to the 5′ UTR of *FAMLF* in Burkitt Lymphomas (BL), allowing the authors to suggest a new mechanism of the pathogenesis of BL, and to propose miR-181b as a candidate therapeutic agent in BL forms with *FAMLF* overexpression [62].

#### 4.2.3. Multiple Myeloma (MM)

In a recent study, the role of miR-181a in MM has been reported [105]. In particular, the authors reported the up-regulation of miR-181a in MM cells and assessed the oncogenic function of miR-181a by showing, in xenograft and MM in vitro models, that overexpression of this miRNA promotes tumor formation both in vitro and in vivo [105].

## 5. Conclusions

The above-reported data demonstrate that the miR-181 family plays relevant roles in fundamental pathophysiological processes. Various studies have shown aberrant expression of these miRNAs in the most common neurodegenerative disorders such as AD and PD as well as in different solid tumors and in hematological malignancies, where they act either as tumor suppressors or oncomirs, highlighting the possibility to use these miRNAs as diagnostic/prognostic biomarkers. 

On the other hand, miR-181 family members should be considered as therapeutic targets for the treatment of various diseases. The reported evidence that their down-regulation protects from neurodegeneration both in vitro and in vivo highlights their therapeutic potential in neurodegenerative conditions. Moreover, further dissection of the pathophysiological mechanisms underlying the miR-181-mediated action in diseased conditions is expected to lead to new therapeutic opportunities for a wide range of tumors where they can be used either alone or in combination with other drugs to increase the efficacy of chemotherapeutic agents.

In conclusion, although more efforts are necessary to better dissect the role of the miR-181 family in different disease contexts, we believe that these miRNAs may represent novel biomarkers for diagnostic and prognostic applications and/or new therapeutic targets in neurodegeneration and cancer.

## Figures and Tables

**Figure 1 ijms-21-02092-f001:**
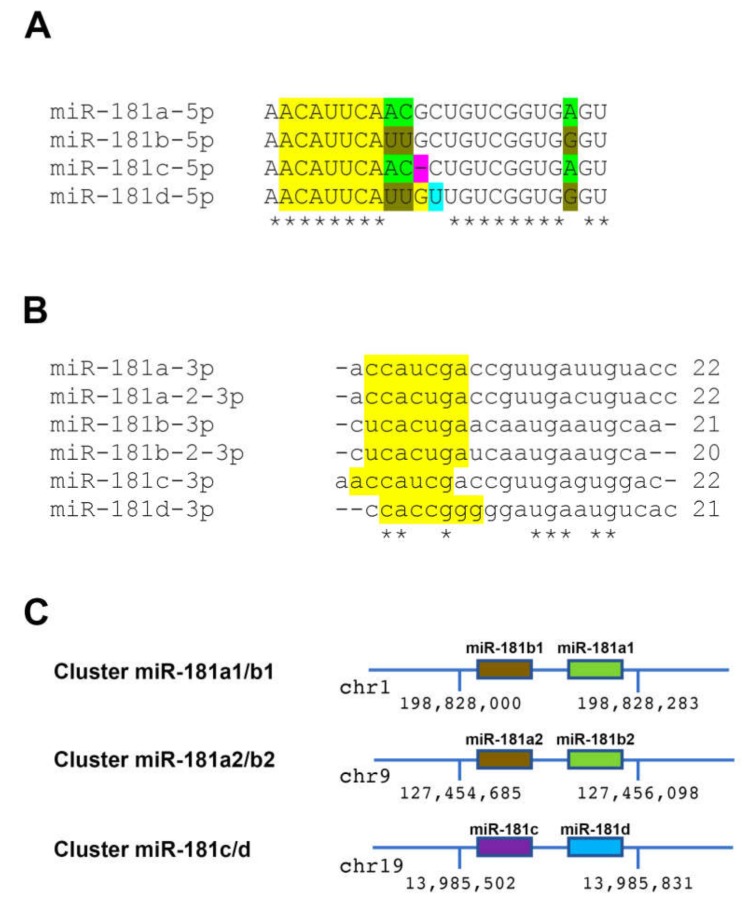
The miR-181 family in the human genome. (**A**) Sequence alignment of the four different 5p mature sequences. The seed is identical in all four 5p members and is labeled in yellow. The three bases that differ between miR-181a and miR-181b are labeled in green and brown, respectively. The single nucleotide positions that constitute the only difference in the miR-181a/c and miR-181b/d pairs are labeled, respectively in purple (in miR-181c) and light blue (in miR-181d). The asterisks mark the identical nucleotides. (**B**) Sequence alignment of the six different 3p mature sequences. The seed regions are labeled in yellow and the asterisks mark the identical bases. (**C**) Genomic organization of the three miR-181 clusters. Chromosomal position and genomic coordinates (hg19) are shown. Please note that the clusters are not drawn to scale.

**Figure 2 ijms-21-02092-f002:**
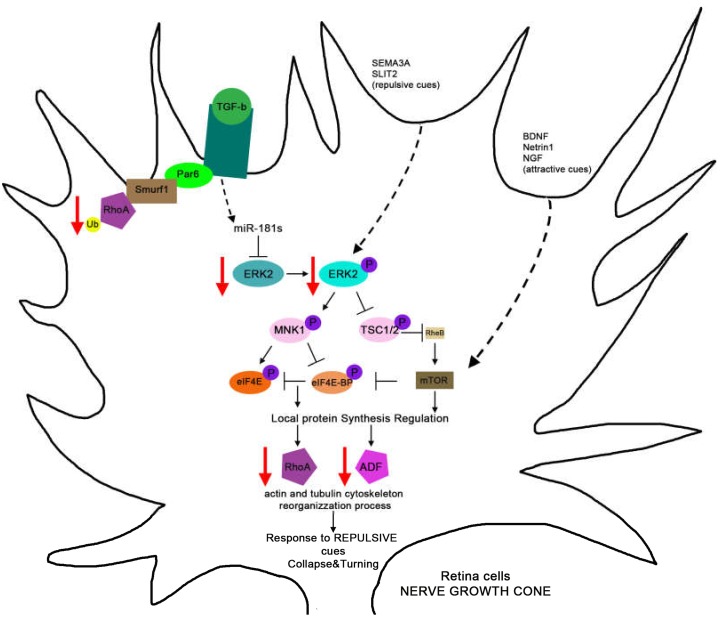
Model for the regulation and function of miR-181a/b in axon specification and growth processes. The TGF-β/MAPK signaling antagonism in the neuronal process formation is exerted via miR-181a/b. The TGF-β signaling regulates, on one hand, the RhoA degradation, and, on the other, the RhoA protein synthesis decrease via regulation of miR-181a/b levels. The miR-181a/b, by regulating the ERK2 levels, can modulate the MAPK signaling cascade and the protein synthesis of RhoA, allowing its physiological down-regulation, needed for neuron polarization and rapid elongation of the neurite. Then during axon elongation, the miR-181a/b effect on ERK2 levels is needed in the modulation of MAPK signaling and Protein synthesis of RhoA and Cofilin (Actin Depolymerization Factor), responsible for cytoskeletal modification involved in axon growth cone collapse and turning.

**Figure 3 ijms-21-02092-f003:**
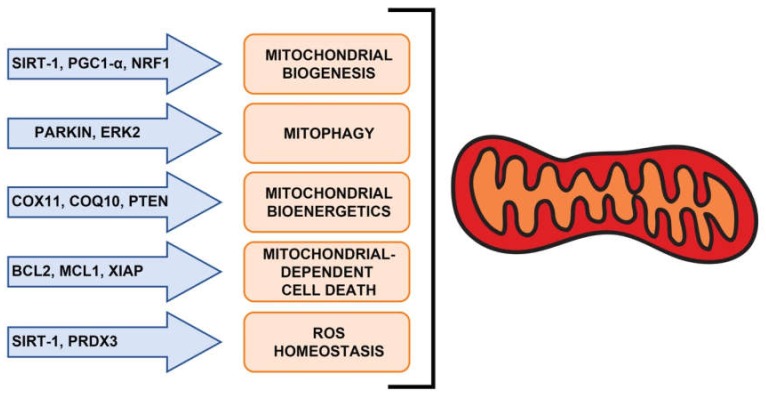
miR-181 family members control pathways important for mitochondrial homeostasis through the targeting of different key players.

**Table 1 ijms-21-02092-t001:** The miR-181 family in development, differentiation, neurodegenerative diseases, and cancer.

**miR-181 Family in Development and Differentiation**	**Early embryo development**	**Role in embryo implantation**	[29]
		Role in gastrulation	[26]
	**Late embryo development**	Role in interdigital tissue remodeling in the embryonic limb	[30]
		Role in chondrocyte development	[31,32,33]
	**CNS development**	Role in neural stem cell self-renewal	[34]
		Retinal axon specification and growth	[26,35]
**miR-181 Family in Neurodegenerative Diseases**	**Alzheimer Disease (AD)**	Altered expression in the brain, blood, and CSF of AD patients	[16,17,18,19,20,21]
		Altered expression in the brain of AD mouse model	[11]
		Protective effect of down-regulation in vivo	[36]
	**Parkinson’s Disease (PD)**	Altered expression in the brain and blood of PD patients	[22,23]
		Direct targeting of *PARK2,* responsible for a familiar form of PD	[37]
	**Mitochondrial Diseases (MDs)**	Protective effect of down-regulation in vivo	[28]
	**Focal cerebral ischemia**	Protective effect of down-regulation in vivo	[38]
**miR-181 Family in Cancer ***	**Solid tumors**	OncomiR or Onco-suppressor role	[39,40,41,42,43,44,45,46,47,48,49,50,51,52,53,54,55,56,57,58,59,60,61,62,63,64,65,66,67,68,69,70,71,72,73,74,75,76,77,78,79,80,81,82,83,84,85,86,87,88,89,90]
	**Hematological cancers**	OncomiR or Onco-suppressor role	[91,92,93,94,95,96,97,98,99,100,101,102,103,104,105]

* See Table 2 for further details.

**Table 2 ijms-21-02092-t002:** The functional role of the miR181 family in cancer.

Cancer	miR-181 Member	Target	Cancer-Related Pathway	Functional Role	miR-181 Expression	Ref.
Hepatocellular carcinoma	miR-181*	CDX2, GATA6	ERK	OncomiR	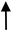	[39]
Hepatocellular carcinoma	miR-181*	NLK	Wnt	OncomiR	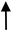	[39]
Hepatocellular carcinoma	miR-181b	TIMP3	TGFb	OncomiR	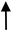	[40]
Hepatocellular carcinoma	miR-181a, -181b, and -181d	MKP-5	MAPK	OncomiR	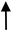	[41]
Colorectal Carcinoma	miR-181a	SRCIN1	VEGF	OncomiR	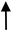	[42]
Colorectal Carcinoma	miR-181b	RASSF1A	EGFR/RAS	Onco-suppressor	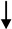	[43]
Colorectal Carcinoma	miR-181a	WIF-1	Wnt	OncomiR	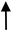	[44].
Non-small cell lung cancer	miR-181*	BCL2	Apoptosis	Onco-suppressor	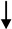	[45]
Non-small cell lung cancer	miR-181b	HMGB1	Apoptosis	Onco-suppressor	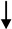	[46]
Non-small cell lung cancer	miR-181a	VCAM-1	NF-κb	Onco-suppressor	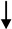	[47]
Pancreatic Cancer	miR-181a	PTEN	PI3K-AKT	OncomiR	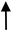	[48]
Pancreatic Cancer	miR-181a	MAP2K4	PI3K-AKT	OncomiR	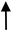	[48]
Pancreatic Cancer	miR-181a	ATM	P53	OncomiR	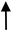	[49]
Ovarian Cancer	miR-181a	SMAD7	TGFb	OncomiR	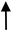	[50]
Ovarian Cancer	miR-181*	RTKN2	NF-κB	Onco-suppressor	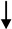	[51]
Prostate cancer	miR-181a	DAX-1	AR	OncomiR	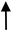	[52]
Prostate cancer	miR-181a	TGIF2	TGFb/ EMT	OncomiR	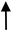	[53]
Breast Cancer	miR-181b	YWHAG	EMT	OncomiR	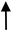	[54]
Breast Cancer	miR-181b	BIM	Apoptosis	OncomiR	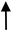	[55]
Breast Cancer	miR-181*	SPRY4	MAPK	OncomiR	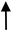	[56]
Brain Cancer/Astrocytoma	miR-181b	NOVA1	Splicing	Onco-suppressor	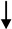	[57]
Brain Cancer/Glioma	miR-181*	Cyclin B1	Cell Cycle	Onco-suppressor	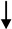	[58]
Brain Cancer/Glioblastoma	miR-181b	KPNA4	NF-κB /EMT	Onco-suppressor	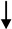	[59]
Brain Cancer/Neuroblastoma	miR-181*	CDON	SHH	OncomiR	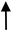	[60]
Brain Cancer/Neuroblastoma	miR-181c	SMAD7	TGFb	Onco-suppressor	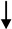	[61]
Brain Cancer/Neuroblastoma	miR-181a	PARK2	Apoptosis	Onco-suppressor	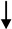	[37]
Haematological Cancers/Lymphoma	miR-181b	FAMLF	Familial acute myelogenous leukemia	Onco-suppressor	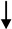	[62]
Haematological Cancers/Leukemia	miR-181b	TCL1	AKT	Onco-suppressor	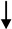	[63]
Haematological Cancers/Leukemia	miR-181b	MCL1	Apoptosis	Onco-suppressor	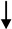	[63]
Haematological Cancers/Leukemia	miR-181b	BCL2	Apoptosis	Onco-suppressor	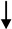	[63]

* specific miR-181 family member not reported.

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
