# Peer review of "The Pervasive Role of the miR-181 Family in Development, Neurodegeneration, and Cancer"

_ijms, 2020, doi:10.3390/ijms21062092_

Round 1
Reviewer 1 Report
In this review article, Alessia Indrieri and colleagues summarized the previous progress and studies to miR-181 family in health and disease. The authors covered comprehensive aspects of miR-181 roles in development, differentiation and diseases, and mentioned lots of related studies with brief descriptions. However, the manuscript is suffering from major and minor issues that may prevent it from publication in the current format. I also found that the article is boring to follow, due to the defects of the way it was written. Under every point, the authors simply listed different relevant studies and without any legible general summary. There is no description of commonness and differences among the results. Basically, the authors have collected sufficient resources, but haven't deeply explained, and discussed. They used of a simple framework of writing “miR-181 family members involve in …… disease that confirmed by ……”. In my opinion, this is the major problem of the article, which makes it boring and difficult to follow.
Here are my comments that might improve the quality of the manuscript.
- The title is so general and “microRNA is redundant”. I think a more specific title could be more informative for readers. Something like “The controversial role of miR-181 family in development, differentiation, neurodegenerative diseases and cancer progression.
- Keywords are also so general. Keywords that are more specific are more helpful.
- Some portions of this study is heavy and is hard to follow. A table that summarized all mentioned phenotypes; development, differentiation, Parkinson, Alzheimer, solid tumors, and leukemia with the short description would be so helpful and could help readers to follow the paper better.
- There are some reports that showed “some miRNAs could upregulate gene expression in specific cell types”. It would be nice if Authors revise the first sentence (line 33) accordingly.
- Although Authors cited 120 references, some References are missing. E.g., Line 38-45, line 74-78, line 99-103 and some sentences/paragraphs cited more than one (For example look at line 37, 85, 147, 231, 234, 236, 237, 265, 382,383)
- Line 46-47 are actually the outline of the manuscript that is poorly written. I recommend highlighting this part with additional description to encourage readers to follow.
- Line 63-73 is vague. I highly recommend rewriting this portion.
- All figures are blurry specially figure 2. I highly recommend making high-resolution figures.
- 2.1 and 2.2 subsections are similar. 2.2 should be late embryo development.
- Line 324-327 is redundant (look at line 548-551). Either rewrite or delete it.
- Figure 3. It would be nice if you add and link all associated miR-181 to the target genes.
- Some abbreviations were not widely used and may be confusing.
- Finally, there are a few typos throughout the manuscript. For example see line 363. The authors should carefully proof the manuscript.
Author Response
Revision changes are highlighted by using the Track Changes in the revised version of the manuscript
Reviewer comments:
In this review article, Alessia Indrieri and colleagues summarized the previous progress and studies to miR-181 family in health and disease. The authors covered comprehensive aspects of miR-181 roles in development, differentiation and diseases, and mentioned lots of related studies with brief descriptions. However, the manuscript is suffering from major and minor issues that may prevent it from publication in the current format. I also found that the article is boring to follow, due to the defects of the way it was written. Under every point, the authors simply listed different relevant studies and without any legible general summary. There is no description of commonness and differences among the results. Basically, the authors have collected sufficient resources, but haven't deeply explained, and discussed. They used of a simple framework of writing “miR-181 family members involve in …… disease that confirmed by ……”. In my opinion, this is the major problem of the article, which makes it boring and difficult to follow.
Here are my comments that might improve the quality of the manuscript.
1. The title is so general and “microRNA is redundant”. I think a more specific title could be more informative for readers. Something like “The controversial role of miR-181 family in development, differentiation, neurodegenerative diseases and cancer progression.
OUR RESPONSE: We thank the reviewer for his/her suggestion and changed the title in “The pervasive role of the miR-181 family in development, neurodegeneration and cancer”.
2. Keywords are also so general. Keywords that are more specific are more helpful.
OUR RESPONSE: We revised the list of keywords.
3. Some portions of this study is heavy and is hard to follow. A table that summarized all mentioned phenotypes; development, differentiation, Parkinson, Alzheimer, solid tumors, and leukemia with the short description would be so helpful and could help readers to follow the paper better.
OUR RESPONSE: As suggested by the Reviewer, we added a summary box that recapitulates the role of the miR-181 family in the described processes, i.e., development, neurodegenerative diseases and cancers.
4. There are some reports that showed “some miRNAs could upregulate gene expression in specific cell types”. It would be nice if Authors revise the first sentence (line 33) accordingly.
OUR RESPONSE: We revised the sentence according to the reviewer’s suggestion (page 1, lines 39-40).
5. Although Authors cited 120 references, some References are missing. E.g., Line 38-45, line 74-78, line 99-103 and some sentences/paragraphs cited more than one (For example look at line 37, 85, 147, 231, 234, 236, 237, 265, 382,383)
OUR RESPONSE: We added the missing references as suggested (page 2, lines 49-56; page 3, lines 93-97; page 5, lines 120-124).
6. Line 46-47 are actually the outline of the manuscript that is poorly written. I recommend highlighting this part with additional description to encourage readers to follow.
OUR RESPONSE: We rewrote this part according to the Reviewer’s suggestion (page 2, lines 57-64).
7. Line 63-73 is vague. I highly recommend rewriting this portion.
OUR RESPONSE: We modified this part according to the reviewer’s suggestion (page 2, lines 80-92).
8. All figures are blurry specially figure 2. I highly recommend making high-resolution figures.
OUR RESPONSE: As requested, we increased the resolution of figures.
9. 2.1 and 2.2 subsections are similar. 2.2 should be late embryo development.
OUR RESPONSE: We fixed this inaccuracy (page 6, line 144).
10. Line 324-327 is redundant (look at line 548-551). Either rewrite or delete it.
OUR RESPONSE: We apologize for this inaccuracy. The former line 324-327 statement was rewritten (page 11, lines 357-361).
11. Figure 3. It would be nice if you add and link all associated miR-181 to the target genes.
OUR RESPONSE: We recognize that it would be ideal to clearly indicate in this Figure which miR-181 family member recognizes which target. However, we believe that this information could be misleading. This is because only in a few cases the capability of ALL of the miR-181 family members to regulate the target under investigation was tested (see for example PMID:21958558 and PMID: 20162574), whereas in some other cases only one member of the family was analyzed. The miR-181 family present high functional redundancy and therefore it is not possible, in the absence of ad-hoc studies, to exclude a promiscuity of targeting capability among the family members. Therefore, for the sake of caution, we prefer not to add this information in Figure 3.
12. Some abbreviations were not widely used and may be confusing.
OUR RESPONSE: We carefully checked and revised all the abbreviations in the text as well as the Abbreviation paragraph.
13. Finally, there are a few typos throughout the manuscript. For example see line 363. The authors should carefully proof the manuscript.
OUR RESPONSE: We carefully proofread the manuscript to remove typos and inaccuracies.
Reviewer 2 Report
Indrieri and colleagues presented an exhaustive review article on the involvement of the miR-181 family in several human pathologies. The authors clearly describe the molecular features of all the members of the miR-181 family describing their physiological and pathological roles. Overall, the manuscript is well structured and contained novel and updated information. However, there are some minor revisions that the authors have to address to improve the quality of the manuscript:
1) In line 36, please use the term “seed region” instead of “seed”. In addition, the authors should describe that the seed region is mostly located at positions 2-7 from the miRNA 5´-end and that it matches completely or partially with the mRNA target leading to the degradation of the mRNA or its translational block, respectively;
2) Please, add references for the following sentence: “Among the so far characterized miRNA families, we will focus on the miR-181 family that has a well-described role in development, mitochondrial function and cancer.”;
3) Figures 1 and 2 are of poor quality. Please provide higher resolution images;
4) In table 1, please add a new column indicating the expression levels of miRNAs (up-regulated or down-regulated);
5) In the paragraph “4. The miR-181 family in Cancer”, the authors state: “Recent findings show the implication of miR-181family members in different solid tumors and in hematological malignancies”. However, Table I does not contain any data about hematological malignancies. Please provide references and information about hematological malignancies and lymphomas and other solid tumors. For this purpose, see:
Lymphoma
- 10.3892/ol.2015.4031
Oral cancer
- 10.3390/cancers11050610
- 10.1111/j.1600-0714.2010.01003.x
Acute Myeloid Leukemia
- 10.3904/kjim.2017.102
- 10.1080/10428194.2016.1272680
- 10.3892/ol.2016.4970
Chronic Myeloid Leukemia
- 10.3892/ol.2015.3663
- 10.3109/10428194.2013.796055
6) The paragraph “3.2. Haematological Cancers” should be better described. Please provide information about both Chronic and Acute Myeloid Leukemia. A wider description of miR-181 family involvement in hematological malignancies is needed.
Author Response
Revision changes are highlighted by using the Track Changes in the revised version of the manuscript
Reviewer comments:
Comments and Suggestions for Authors
Indrieri and colleagues presented an exhaustive review article on the involvement of the miR-181 family in several human pathologies. The authors clearly describe the molecular features of all the members of the miR-181 family describing their physiological and pathological roles. Overall, the manuscript is well structured and contained novel and updated information. However, there are some minor revisions that the authors have to address to improve the quality of the manuscript:
1) In line 36, please use the term “seed region” instead of “seed”. In addition, the authors should describe that the seed region is mostly located at positions 2-7 from the miRNA 5´-end and that it matches completely or partially with the mRNA target leading to the degradation of the mRNA or its translational block, respectively;
OUR RESPONSE: We implemented the reviewer’s suggestion (pages 1-2, lines 42-48).
2) Please, add references for the following sentence: “Among the so far characterized miRNA families, we will focus on the miR-181 family that has a well-described role in development, mitochondrial function and cancer.”;
OUR RESPONSE: We added the missing references and we also expanded this section to comply with Reviewer 1’s suggestion (page 2, lines 57-64).
3) Figures 1 and 2 are of poor quality. Please provide higher resolution images;
OUR RESPONSE: we increased the resolution of Figures 1 and 2.
4) In table 1, please add a new column indicating the expression levels of miRNAs (up-regulated or down-regulated);
OUR RESPONSE: We added a column displaying the expression changes of miR-181 family members in the described condition.
5) In the paragraph “4. The miR-181 family in Cancer”, the authors state: “Recent findings show the implication of miR-181family members in different solid tumors and in hematological malignancies”. However, Table I does not contain any data about hematological malignancies. Please provide references and information about hematological malignancies and lymphomas and other solid tumors. For this purpose, see:
Lymphoma
- 10.3892/ol.2015.4031
Oral cancer
- 10.3390/cancers11050610
- 10.1111/j.1600-0714.2010.01003.x
Acute Myeloid Leukemia
- 10.3904/kjim.2017.102
- 10.1080/10428194.2016.1272680
- 10.3892/ol.2016.4970
Chronic Myeloid Leukemia
- 10.3892/ol.2015.3663
- 10.3109/10428194.2013.796055
OUR RESPONSE: We are grateful to the Reviewer for spotting the missing data. We revised Table 1 as requested including references and information about hematological malignancies and lymphomas and other solid tumors.
6) The paragraph “3.2. Haematological Cancers” should be better described. Please provide information about both Chronic and Acute Myeloid Leukemia. A wider description of miR-181 family involvement in hematological malignancies is needed.
OUR RESPONSE: We revised paragraph 3.2 and provided the missing information on Chronic and Acute Myeloid Leukemia. Moreover, an introductory paragraph about the role of miR-181 family in Hematological Cancers has been added (pages 17-18, lines 559-614).
Round 2
Reviewer 1 Report
Most of my concerns from my previous review have been addressed and I would be happy to recommend acceptance.